# The Pro-Gly or Hyp-Gly Containing Peptides from Absorbates of Fish Skin Collagen Hydrolysates Inhibit Platelet Aggregation and Target P_2_Y_12_ Receptor by Molecular Docking

**DOI:** 10.3390/foods10071553

**Published:** 2021-07-05

**Authors:** Qi Tian, Shi-Ming Li, Bo Li

**Affiliations:** 1College of Food Science and Nutritional Engineering, China Agricultural University, Beijing 100083, China; tian.q@foxmail.com (Q.T.); shiming.li@cau.edu.cn (S.-M.L.); 2Key Laboratory of Functional Dairy, Ministry of Education, Beijing 100083, China

**Keywords:** antiplatelet, collagen hydrolysates, Hyp-Gly, molecular docking, P_2_Y_12_ receptor

## Abstract

Previous studies found that the collagen hydrolysates of fish skin have antiplatelet activity, but this component remained unknown. In this study, eleven peptides were isolated and identified in the absorbates of Alcalase-hydrolysates and Protamex^®^-hydrolysates of skin collagen of *H. Molitrix* by reverse-phase C18 column and HPLC-MS/MS. Nine of them contained a Pro-Gly (PG) or Hyp-Gly (OG) sequence and significantly inhibited ADP-induced platelet aggregation in vitro, which suggested that the PG(OG) sequence is the core sequence of collagen peptides with antiplatelet activity. Among them, OGSA has the strongest inhibiting activities against ADP-induced platelet aggregation in vitro (IC_50_ = 0.63 mM), and OGSA inhibited the thrombus formation in rats at a dose of 200 μM/kg.bw with no risk of bleeding. The molecular docking results implied that the OG-containing peptides might target the P_2_Y_12_ receptor and form hydrogen bonds with the key sites Cys97, Ser101, and Lys179. As the sequence PG(OG) is abundant in the collagen amino acid sequence of *H. Molitrix,* the collagen hydrolysates of *H. Molitrix* might have great potential for being developed as dietary supplements to prevent cardiovascular diseases in the future.

## 1. Introduction

In the last decade, with the increase in the aging population, the incidence and mortality of cardiocerebrovascular diseases (CCVd) are still on the rise in China. Researchers predict that there are 290 million CCVd patients in China and believe that the disease has become the primary threat to the health of the elderly [1]. Thrombus formation is the pathological basis of CCVd, and its key factors are activation, aggregation, and adhesion of platelets [2,3]. Platelet activation is a process in which platelets in a resting state release particles, adhere, and aggregate under the stimulation of activating agents such as collagen [4], ADP [5], and thrombin [6,7]. Platelet activation plays a decisive role in physiological hemostasis. However, excessive activation of platelets is involved in many pathological processes, such as atherosclerosis, thrombosis, tumor metastasis, and inflammation [8]. In recent years, a variety of potent antiplatelet drugs have been designed to inhibit platelet aggregation, such as thromboxane A2 (TXA2) inhibitor aspirin [9], P_2_Y_12_ receptor antagonists clopidogrel, and prasugrel [10]. However, long-term antiplatelet drug treatment may have a variety of side effects, including bleeding, gastrointestinal damage, allergic reactions, headaches, and liver damage [11,12].

Compared with antiplatelet drugs, preventing and suppressing thrombotic diseases through diets have unique advantages: fewer side effects and less cost. At present, a large number of studies reported that various natural antiplatelet compounds in food can inhibit platelet aggregation, including coumarins from the plant family *Rutaceae* [13], mangiferin [14], polyphenols from tomato [15], resveratrol [16], and quercetin [17].

Collagen is the structural protein of the extracellular matrix. It is abundant in skin, bone, and cartilage tissues, accounting for about 30% of the total protein in animals [18]. In recent years, the separation and identification of bioactive peptides from collagen hydrolysates, such as antioxidant peptides [19,20], anti-inflammatory peptides [21], and antiosteoporosis peptides [22], has attracted the attention of researchers. Previous studies reported that the peptides, such as DGEA, FGN, and GPR, derived from type I collagen in pig skin or fish skin have antiplatelet activity [23,24,25]. In our previous study, thirteen-month-old mice were administered with collagen hydrolysates from *H. Molitrix* skin for 2 months. The collagen hydrolysates significantly decreased the level of platelet release indicators in the plasma, including PF4, granule membrane protein (GMP)-140, -thromboglobulin, and serotonin [26]. Later, several antiplatelet peptides were identified from *H. Molitrix* collagen hydrolysate after simulated gastrointestinal digestion and intestinal absorption, including OG, OGE, PGEOG, and VGPOGPA, one of which, OGE, was also confirmed to inhibit thrombosis in vivo [27].

Molecular docking is a computer-based structure simulation method that is widely used in drug discovery and development. Since its first appearance in the mid-1970s, it has proven to be an important tool to help reveal how a compound interacts with its target protein [28]. In this study, molecular docking was used to reveal the target receptor proteins and binding sites and the bonds formed by their ligand-target interactions.

In short, the purpose of this study was to identify the antiplatelet peptides from the absorbates of Alcalase-hydrolysates and Protamex^®^-hydrolysates of the skin collagen of *H. Molitrix* and assess their antiplatelet activity. Furthermore, the target receptors of these peptides on platelets were simulatively screened by molecular docking.

## 2. Materials and Methods

### 2.1. Material and Chemicals

*H. Molitrix* skin was supplied by Hubei Zhongke Agriculture Co., Ltd. (Jingzhou, China). Adenosine diphosphate (ADP), trypsin -EDTA solution was purchased from Solarbio (Beijing, China). Alcalase and Protamex^®^ were purchased from Novozymes (Beijing, China). Collagen type I from rat tail tendon, pancreatin from porcine, and pepsin from porcine gastric mucosa were purchased from Sigma-Aldrich (Shanghai, China). Phosphate buffer solution (PBS) was purchased from HyClone (Beijing, China). Dulbecco modified Eagle’s minimal essential medium (DMEM), Fetal bovine serum (FBS), Hank’s buffered saline solution (HBSS), nonessential amino acids (NEAA), penicillin, and streptomycin were purchased from Gibco (Life Technologies, Grand Island, NY, USA). Sodium hydroxide (NaOH), trisodium citrate (Na_3_C_6_H_5_O_7_·2H_2_O), hydrochloric acid (HCl), sodium bicarbonate (NaHCO_3_), and other chemicals were all analytically pure grade and purchased from Sinopharm chemical reagent Co., LTD (Beijing, China).

### 2.2. Preparation of Collagen Peptides (CPs)

Gelatin from the *H. Molitrix* skin was obtained according to the previously described method [26]. The gelatin was enzymatically hydrolyzed by Alcalase for 4.0 h (AH), or Protamex^®^ for 8.0 h (PH).

### 2.3. Simulated Gastrointestinal Digestion

The in vitro gastrointestinal digestion of collagen peptide has been described in a previous report with a minor modification [27]. During simulated gastric digestion, pepsin (a ratio of enzyme to substrate of 1:50, *w*/*w*, and pH 2.0) was added to CPs (a concentration of 4%, *w*/*v*) and incubated at 37 °C for 2 h. During intestinal digestion, the mixture from simulated gastric digestion was adjusted to pH 7.0 with 2 M NaOH. Pancreatin (1:50, enzyme/CPs ratio, *w*/*w*) was further added to hydrolyze the mixture at 37 °C for 2 h and was stirred continually during the digestion progress. Finally, the mixture was heated in boiling water for 10 min to inactivate the enzymes. The digested solution was centrifuged at 8000 rpm for 15 min. Finally, the supernatant was freeze dried and stored at −80 °C in the dark for further Caco-2 treatment.

### 2.4. Uptake and Transport by Caco-2 Cells

The Caco-2 cell culture method has been described in a previous report [29]. Caco-2 cells were seeded and cultivated in 25 cm^2^ plates, and the cells were incubated in an atmosphere of 5% CO_2_ with DMEM containing 10% FBS, 1% nonessential amino acids, 100 U/mL penicillin, and 0.1 mg/mL streptomycin.

When Caco-2 cells covered 80% to 90% of the culture area of the flask, cells were dislodged using trypsin. Then, Caco-2 cells were seeded and cultivated at a density of 1 × 10^5^/mL on the upper side of the 6-well Transwell cell culture plate (AP). The medium was replaced every 2 days. After 21 days of incubation, a transepithelial electrical resistance (TEER) value of 800 Ω cm^2^ or higher indicated that the cells were differentiated in monolayers [30].

Lyophilized polypeptide samples were dissolved in HBSS at a dose of 15 mg/mL. The monolayer membrane was gently rinsed with preheated HBSS for 2–3 times, 1.5 mL HBSS solution was added to the upper (AP) side of the cell model, and 3 mL HBSS solution was added to the lower (BL) side. Then, the cell model was placed in a cell incubator and balanced at 37 °C for 30 min. After that, the solution on the AP side was removed and 1.5 mL sample was added. The sampled plates were placed in a cell incubator for 2 h. The solution on the BL side was collected for all experiments below.

### 2.5. Separation of the AH Absorbate (AHA) and PH Absorbate (PHA)

Each absorbate was desalted through a C18-SPE column (ODS-A, YMC Co., Kyoto, Japan) before separation. The absorbate was dissolved with an appropriate amount of deionized water and loaded on the column. Then, the column was washed with deionized water, 50% ethanol aqueous solution. The elution flow rate was 1.0 mL/min, and detection wavelength was 220 nm. The peptide components were collected after 10 min.

After desalination, the sample was loaded on the ODS-A universal inverse-phase C18 column for separation. Elution conditions are as follows: 0–15 min, distilled water; 15–25 min, 10% methanol; 25–35 min, 30% methanol; 35–45 min, 50% methanol. The flow rate was 1.0 mL/min, and the detection wavelength was 220 nm. The HD-A chromatographic processing system was used to record the atlas and collect the fractions. Each fraction was concentrated by rotary evaporation at 45 °C then freeze dried and stored at −80 °C.

### 2.6. Peptide Sequencing Using LC-MS/MS

The peptides were identified by capillary high-performance liquid chromatography-electrospray ionization tandem mass spectrometry (HPLC-ESI-MS/MS) analysis using the Thermo Fisher Scientific™ HPLC (Ultimate 3000) coupled to an Orbitrap Elite™ Hybrid Ion Trap–Orbitrap Mass Spectrometer (Thermo Fisher Scientific (China) Co., Ltd., Shanghai, China) according to previously reported methods [29]. The mass spectrometer was operated in the positive mode with the capillary voltage of 3.6 kV and the source temperature of 100 °C. The scanning m/z range was 200–1500 in the MS mode and 50–1500 in the MS/MS mode. The MS/MS data were processed using Thermo Xcalibur software (Thermo Fisher Scientific, San Jose, CA, USA) in combination with manual de novo sequencing. The spectra of hydrolysates were processed to MS (mass spectrum) and MS/MS, respectively, by Bruker Daltonics Data Analysis 4.0 (Bruker Daltonics Inc., Billerica, MA, USA). Analysis results were confirmed using protein database from National Center for Biotechnology Information (NCBI, https://www.ncbi.nlm.nih.gov/protein, accessed on 17 September 2018). Validated by Peptide/Protein MS Utility program (http://prospector.ucsf.edu/, accessed on 20 September 2018). Peptides were synthesized using the method of Fmoc solid-phase (GL Biochem Ltd., Beijing, China). The purity of all peptides used in the study was greater than 98% determined by HPLC.

### 2.7. Antiplatelet Activity Assay In Vitro

Blood was taken from the abdominal aorta of healthy male SD rats and mixed with 3.8% sodium citrate (anticoagulant: blood = 1: 9, *v*/*v*). Then, the solution with equal volume of PBS was centrifuged at 50× *g* for 10 min at 23 °C. The supernatant was centrifuged under the same conditions again. After that, supernatant was centrifuged at 750× *g* for 10 min at 23 °C. The upper layer was poor platelet plasma (PPP), and the precipitation was platelets. Platelets were suspended by plasma, and platelet concentrations were adjusted to 2–3 × 10^8^/mL to obtain platelets-rich plasma (PRP).

A measurement of 270 μL PRP was placed in the test cup and preincubated at 37 °C for 5 min. For the model induced by collagen or thrombin, 30 μL of Tyrode’s buffer (model group), peptides (2, 4 mg/mL, sample groups), tripeptide RGD(2, 4 mg/mL, positive control group 1), and aspirin (1.5 mM, positive control group 2) were added to the test cup and incubated at 37 °C for 5 min, respectively. Then, the activators collagen (50 μg/mL) or thrombin (0.5 U/mL) were added, respectively, to measure the platelet aggregation rate at 300 s. For ADP-induced model, 30 μL PPP (model group), peptides (sample group), tripeptide RGD, and aspirin (positive control group) were added to the test cup and incubated at 37 °C for 5 min. Finally, the activators ADP (100 μM) were added to measure the platelet aggregation rate at 300 s [31].

### 2.8. Animals, Diets, and Treatments

#### 2.8.1. Feeding Conditions and Animal

All procedures involving experimental animals were performed in accordance with the protocols approved by the Committee for Animal Research of Peking University (No.KY150018) and conformed to the Guide for the Care and Use of Laboratory Animals (NIH publication No.86-23, revised 1996). The experiment was approved by the Animal Experimental Welfare and Ethical Inspection Committee, the Supervision, Inspection and Testing Center of Genetically Modified Organisms, Ministry of Agriculture (Beijing, China). Eight-week-old male SD rats (180 ± 5 g, SPF grade) were purchased from Beijing Vital River Experiment Animal Technology Co., Ltd. (Beijing, China). All rats were housed in cages (6–8 rats per cage) and were allowed free access to normal diet and water. Animal room was maintained at a temperature of 23 ± 2 °C, relative humidity of 50 ± 5%, and artificially illuminated with a 12 h light/dark cycle. All rats were allowed a normal diet and water freely. After adapting to the environment for 6 days, rats were grouped based on bodyweight into four groups (*n* = 5/group), including the normal group (N, Normal saline), high dose peptides group (HP, 300 μmol/kg.bw), low dose peptides group (LP, 200 μmol/kg.bw), and positive control group (Clopidogrel, 100 μmol/kg.bw). All groups were orally administrated.

#### 2.8.2. Ferric Chloride-Induced Arterial Thrombosis Model

After an hour, rats were anesthetized with 2% sodium pentobarbital (0.25 mL/100 g.bw). The neck skin was removed, and the muscle layer was cut open along the midline. Avoiding lymphatic and glands, the left carotid artery was separated about 1 cm. The carotid artery was encircled with a filter paper strip (1 cm × 0.5 cm) soaked with 10% FeCl_3_ solution, then the wound was covered with gauze soaked with normal saline and kept for 15 min. After that, the filter paper strip was removed and kept for 40 min. After clamping the two ends of the blood vessel with hemostatic tweezers, blood was taken from the abdominal aorta [32,33]. After blood collection, the blood was mixed with 3.8% sodium citrate (9:1, *v*/*v*) for anticoagulation. After gently mixed, the blood was centrifuged for 15 min at 1500× *g*. The upper light-yellow plasma was collected and stored at −80 °C for later coagulation function test. The blood vessel segment was cut off, about 1 cm wrapped in filter paper strip, the residual blood inside was absorbed with filter papers, the overall wet weight was weighed, and the blood vessel was weighed after taking out the thrombus. After the rats were sacrificed, the thymuses and spleens were collected and weighed (mg) to obtain the thymus index and spleen index.

#### 2.8.3. Bleeding Assay and Determination of Thymus Index and Spleen Index

After blood collection, the blood was mixed with 3.8% sodium citrate (9:1, *v*/*v*) for anticoagulation. After being gently mixed, the blood was centrifuged for 15 min at 1500× *g*. The upper plasma was collected and stored at −80 °C. APTT and PT assay: 130 μL rat plasma was put into Automated Coagulation Analyzer (HEMAVET950, Erba Diagnostics Inc., London, UK), then the values of PT and APTT were obtained from the instrument.

TT assay: 200 μL rat plasma was set at 37 °C for 5 min. Then, 200 μL TT reagent was added to the system to record the solidification time.

After the rats were sacrificed, the thymus and spleen were removed and weighed to obtain the thymus index and spleen index.

#### 2.8.4. Determination of Thymus Index and Spleen Index

After the rats were sacrificed, the thymuses and spleens were collected and weighed (mg) to obtain the thymus index (TI) and spleen index (SI). The TI and SI were calculated according to the following equation: TI/SI (mg/g) = (weight of thymus/spleen)/body weight.

### 2.9. Molecular Docking

The interaction mode between antiplatelet peptides and receptors (P_2_Y_12_, PLC β2, PLC β3) was evaluated using molecular docking. The 2D structures of peptides were prepared by ChemBioDraw ultra 14.0 (Perkinelmer co., ltd., Waltham, MA, USA) and converted to 3D format by ChemBio3D ultra 14.0 (Cambridge Soft Corporation, USA). The X-ray crystal structures of P_2_Y_12_ (PDB ID:4PXZ), PLC β2 (PDB ID:2ZKM) and PLC β3 (PDB ID:4GNK) were downloaded from the RCSB protein database. Changes such as adding polar hydrogen and water molecule deletion were done through modules of Sybyl-X 2.0 software (Tripos Software, Inc., Berkeley, CA, USA). The docking simulating was done by the triangle matcher placement algorithm plus ΔG_bind_ score and force field as refinement process. The top-score docking poses were chosen for final ligand-target interactions analysis via the Surflex-Dock module of Sybyl-X 2.0 [34,35,36].

### 2.10. Statistics

All data were shown as mean ± SD, calculated, and statistically analyzed (one-way ANOVA, *n* = 5, * *p* < 0.05, ** *p* < 0.01) using SPSS 23.0 software (Microsoft Corp., Washington, DC, USA) and GraphPad 6.0 (GraphPad Inc., San Diego, CA, USA) software packages.

## 3. Results

### 3.1. Effect of Alcalase-Hydrolysates (AH) and Protamex^®^-hydrolysates (PH) on Inhibiting Platelet Aggregation Induced by Collagen, Thrombin, and ADP In Vitro

Alcalase protease and Protamex^®^ protease were used to hydrolyze the skin collagen of *H. Molitrix* to obtain different hydrolysates (AH and PH). The antiplatelet activity of the hydrolysates in the platelet aggregation model of the three inducers (collagen, thrombin, and ADP) was evaluated, respectively. According to the results (Figure 1), the AH significantly inhibited the platelet aggregation caused by the three different inducers (** *p* < 0.01). The PH only had a significant antiplatelet activity in the ADP-induced platelet aggregation model (** *p* < 0.01).

### 3.2. Separation and Identification of Antiplatelet Peptides from Collagen Hydrolysates after Simulated Gastrointestinal Digestion and Intestinal Absorption

After simulated gastrointestinal digestion and intestinal absorption, the absorbates of collagen hydrolysate were separated to identify the antiplatelet peptides. Four fractions were isolated from the AHA and PHA, respectively, named A1, A2, A3, and A4, and P1, P2, P3, and P4, as shown in Figure 2A,B (AHA/PHA: the absorbate of collagen hydrolysate prepared by Alcalase/Protamex^®^). At a dose of 4 mg/mL, all four fractions of AHA had a significant inhibitory effect on ADP-induced platelet aggregation, and the fraction A1 had the highest inhibitory rate. For PHA, fractions P2 and P3 had a significant inhibitory effect on ADP-induced platelet aggregation, and P3 had the highest inhibitory rate (Figure 2C). Then, the fractions with high bioactivity (A1, A4 and P3) were used for further identification of antiplatelet peptides by HPLC-ESI-MS/MS.

A total of 11 peptides from A1, A4, and P3 were identified by amino acid sequencing: A1 (OG, OGSA, PGPK, PGKP, PGPQ, PGQP, GTOGT, EGPAGPA), A4 (OGOMG, VVGOKG), and P3 (PGHH). Among them, the primary mass spectrum and secondary mass spectrum of OGSA in A1, OGOMG in A4, and PGHH in P3 were illuminated in Figure 3.

### 3.3. Effect of Identified Peptides on Inhibiting Platelet Aggregation Induced by Collagen, Thrombin, and ADP In Vitro

Furthermore, 11 identified peptides were synthesized with Fmoc solid-phase. According to platelet aggregation inhibition results (Table 1 and Appendix A), in the collagen-induced model, only the PGPQ showed a decent platelet aggregation inhibition rate (65.6 ± 14.1%); the others have poor inhibition effects or adverse effects. For the thrombin-induced model, the EGPAGPA showed a decent platelet aggregation inhibition rate (53.9 ± 2.1%), while the others have no or weak inhibition effects. Interestingly, the 11 collagen peptides identified all showed high inhibition rates of platelet aggregation in the ADP-induced model, including four the OG-containing peptides OG, OGSA, GTOGT, and OGOMG; five PG-containing peptides PGPK, PGKP, PGPQ, PGQP, and PGHH; and EGPAGPA and VVGOKG. The results suggested that these identified antiplatelet peptides inhibit platelet aggregation through the ADP pathway. Moreover, the OG-containing peptides commonly have a higher antiplatelet activity than the PG-containing peptides. The peptide with the highest antiplatelet activity was the tetrapeptide OGSA, and its IC_50_ was 0.63 mM, which was significantly lower than the dipeptide OG (1.42 mM) (Figure 4A).

### 3.4. Tetrapeptide OGSA Inhibited Thrombus Formation In Vivo

OGSA was selected as an example to evaluate its antithrombotic effect in vivo due to its outstanding effect on inhibiting platelet aggregation in vitro. The FeCl_3_-induced arterial thrombosis model in rats was used to evaluate the antithrombotic activity of OGSA. OGSA could inhibit the thrombus formation significantly at doses of 200 and 300 μM/kg body weight (bw), respectively (Figure 4B). The thrombus weight of the high dose OGSA group was similar to that of the clopidogrel-treated group (93 μM/kg.bw, positive control).

The side effects (such as bleeding risk and immune response) of OGSA were further evaluated in vivo [37]. The prothrombin time (PT), activated partial thromboplastin time (APTT), and thrombin time (TT) in rats after oral administration of OGSA were determined to assess its potential bleeding risk. The results showed that OGSA (200 and 300 μM/kg.bw.) did not affect PT, APTT, or PT significantly (Figure 4C), but the clopidogrel (93 μM/kg.bw) significantly increased TT (*p* < 0.01). The spleen index (SI) and thymus index (TI) of all rats were determined. Neither low dose (200 μM/kg.bw) nor high dose (300 μM/kg.bw) significantly affected TI and SI (Appendix A), indicating that OGSA had few side effects on immune function in rats.

### 3.5. Molecular Docking of Antiplatelet Peptides with P_2_Y_12_ Receptor

Molecular docking is a simulation method that can help researchers reveal the target receptor proteins and binding sites and the bonds formed by their ligand-target interactions [38]. In this paper, we used protein and ligand-based virtual screening techniques for identifying the target protein by comparative evaluation.

The results of inhibiting platelet aggregation in vitro suggested that the collagen peptides identified inhibit platelet aggregation through the ADP activated signal pathway. The receptor protein of ADP (P_2_Y_12_) and the nodal protein in the signaling pathway (PLC β2 and PLC β3) were molecularly docked with PG(OG) peptides for identifying the receptor protein. Nine antiplatelet peptides mentioned above and four antiplatelet peptides (OGE, PGEOG, VGPOGP, and PGE) containing the PG(OG) sequence from previous studies [27] were used for molecular docking through Auto Dock software. The binding free energy (ΔG_bind_) is usually used to describe the interaction between the ligands and receptors, and its value is negatively correlated with the strength of the binding [39]. The ΔG_bind_ binding energy with P_2_Y_12_, PLC β2, and PLC β3 was shown in Table 2. The smaller the ΔG_bind_, the stronger the binding between the receptor and the ligand. The correlation analysis of ΔG_bind_ and antiplatelet activities (IC_50_ values) was carried out through Pearson correlation analysis; the result showed that the correlativity between the ΔG_bind_ of peptides docking to the P_2_Y_12_ receptor and the antiplatelet activities was the strongest (Pearson correlation coefficient, 0.802). From this result we speculated that the antiplatelet peptides containing the PG(OG) sequence might target the P_2_Y_12_ receptor, which should be confirmed by further experiments.

The possible binding pocket of peptides (Figure 5A) was determined on the surface of P_2_Y_12_ receptor using the Multi-Channel Surfaces module of Sybyl-X 2.0 software (Tripos Corporation, USA). The results showed that the Pearson correlation coefficient between the ΔG_bind_ of these peptides binding with pocket 1 and their antiplatelet activities (IC_50_ values) was the largest (0.612), suggesting that the peptides containing the PG(OG) sequence may bind to the P_2_Y_12_ receptor in its binding pocket 1 (Figure 5B).

The molecular docking scores of 13 antiplatelet peptides containing PG(OG) sequences with the P_2_Y_12_ receptor were shown in Appendix A. The docking findings were mainly evaluated by the total score and crash and polar values. The total score of 12 peptides was higher than 6.00, indicating that their binding is stable. OGE has the highest antiplatelet activity among the 13 peptides (IC_50_ = 0.62 mM) and the lowest ΔG_bind_ with the P_2_Y_12_ receptor (−6.62 kcal/mol) (Table 2). The docking findings showed that OGSA could bind to the receptor via H-bond interaction with the amino acid residues Asn159, Cys97, Ser101, Arg93, and Lys179 in the receptor pocket 1 (Figure 5C,D). OGE could bind to the receptor via H-bond interaction with the amino acid residues Met152, Cys97, Ser101, Asn191, and Lys179 (Figure 5E,F). In addition, the molecular docking results of PGPQ, which has lower antiplatelet activity in vitro (IC_50_ = 3.56 mM), were also illuminated (Figure 3G,H). The PGPQ could bind to the P_2_Y_12_ receptor via H-bonds with the amino acid residues Arg 93, Cys175, Ser101, Asn191, Tyr259, and Lys280. For the molecular docking results of all 13 peptides, it is worth noting that PG(OG)-containing peptides with stronger antiplatelet activity form hydrogen bonds with the sites Cys97, Ser101, and Lys179 in pocket 1 of P_2_Y_12_; thus, the activity of these antiplatelet peptides might depend on the key sites Cys97, Ser101, and Lys179.

## 4. Discussion

Previous experiments showed that the collagen hydrolysate treated with mixed enzymes (Papain, Trypsin, Alcalase) after simulated gastrointestinal digestion and intestinal absorption can significantly inhibit platelet aggregation [27]. In this study, Alcalase and Protamex^®^ were respectively used to prepare collagen hydrolysates instead of the previous mixed enzymes. Alcalase-hydrolysates (AH) has antiplatelet activity under three induction conditions (collagen, thrombin, ADP) in vitro, and Protamex^®^-hydrolysates (PH) only inhibited ADP-induced platelet aggregation, indicating that collagen hydrolysates prepared by different enzymes have different bioactive compounds. AH has great potential compared to PH for being developed as a dietary supplement enriched in antiplatelet peptides. Ten antiplatelet peptides were identified from the absorbate of AH. It is worth noting that 9 of 11 identified antiplatelet peptides contain the PG(OG) sequence. Considering the other four OG-containing antiplatelet peptides reported previously [27], we speculated that the PG(OG)-containing collagen peptides may commonly have antiplatelet activity in vitro.

In the platelet aggregation inhibition assay of the peptides identified, the results based on different inducer models (collagen, thrombin, ADP) showed that the inhibition rate of all 11 peptides in the ADP-induced model (39.8 ± 1.9% to 96.4 ± 2.0%) was significantly higher than that of the collagen-induced or thrombin-induced model. In the latter two models, only a few peptides inhibited platelet aggregation in vitro, and some peptides even showed the opposite effect. Therefore, the antiplatelet molecular mechanism of these peptides may be related to the signaling pathway by which ADP activates platelets. After the platelet release response is activated, the ADP stored in the platelets will be immediately released and activate other platelets. The platelet surface receptors P_2_Y_1_ are rare in rats, thus the key membrane receptor on the ADP pathway that cause platelet aggregation is P_2_Y_12_ [40]. After activation of the P_2_Y_12_ receptor, the Gi protein bound to the P_2_Y_12_ receptor started to inhibit adenylate cyclase (AC), resulting in a decrease in the level of cyclic adenosine monophosphate (cAMP). This would dephosphorylate the phosphorylation of vasodilation-stimulating phosphoprotein (VASP), which causes platelet aggregation [4]. The Gq protein bound to the receptors P_2_Y_1,_ PAR1 (thrombin receptor), and TP (TXA2 receptor), which activate the Phospholipase C (PLC) β family (the Gq-PLCβ signaling pathway). The Gq-PLCβ leads to an increase in the level of Ca^2+^, which also causes platelet aggregation [40]. Thus, the membrane surface receptor P_2_Y_12_ and the intramembrane receptors PLC β2 and PLC β3 were selected as candidates for identifying the target protein by molecular docking with the antiplatelet peptides containing the PG(OG) sequence. As the antiplatelet activity of peptides depends on their binding capacity of the target protein and peptides, we identified the target protein according to the Pearson correlation coefficient, which was the correlation analysis between the ΔG_bind_ of these peptides binding with the candidate protein and their antiplatelet activities. The results indicated that the antiplatelet activity of peptides was highly related to their binding capacity of the peptides and P_2_Y_12_ receptor (Pearson correlation coefficient was 0.802), suggesting that the P_2_Y_12_ receptor was the target protein of the PG(OG)-containing peptides. Furthermore, by comparing their binding sites in pocket 1 of P_2_Y_12_, we found that the key binding sites responsible for the recognition of the PG(OG) peptides might be Cys 97, Lys 179, and Ser 101.

In a previous molecular docking study on ADP inducer-2MeSADP [41], the hydroxyl group on the core structure of the compound formed hydrogen bonds with five sites (Cys 97, Lys 179, Thr 163, Asn 159, and His 187) in the pocket 1 of the P_2_Y_12_ receptor. Two binding sites (Cys 97 and Lys 179) are the same as the binding sites of the (PG)OG-containing peptides with the P_2_Y_12_ receptor. Then, we speculated that the OG-containing peptides may inhibit the binding of ADP to the P_2_Y_12_ receptor in pocket 1 through competitive inhibition, thereby inhibiting ADP-induced platelet aggregation.

Moreover, the sites of identified peptides in the collagen amino acid sequence of *Hypophthalmichthys molitrix* were listed in Appendix A and Appendix A (Hyp is the hydroxylated Pro). The identified sequence PG is especially abundant in the skin collagen of the *Hypophthalmichthys Molitrix*. There are 117 and 99 PG sequences in the α1 and α2 chains of type I collagen, respectively. Therefore, the collagen hydrolysates from *H. Molitrix* might have great potential for developing dietary supplements to prevent cardiovascular diseases in the future.

## 5. Conclusions

In conclusion, this study found that AH has great potential compared to PH for developing dietary supplements enriched in antiplatelet peptides. Ten antiplatelet peptides were identified from the absorbate of AH, and nine peptides contained the PG(OG) sequence. Additionally, all PG(OG)-containing collagen peptides inhibited platelet aggregation in vitro, suggested that the PG(OG)-containing peptides may commonly have antiplatelet activity. Furthermore, the target protein of the PG(OG)-containing antiplatelet peptides was identified as the P_2_Y_12_ receptor by molecular docking. The key binding sites in pocket 1 of the P_2_Y_12_ receptor with PG(OG) might be Cys97, Ser101, and Lys179, which may contribute to the antiplatelet activity of these peptides. In addition, the identified sequence PG(OG) is abundant in skin collagen of the *H. Molitrix*. These newly found insights highlight the potential application of the hydrolysates of skin collagen from *H. Molitrix* as dietary supplements to prevent cardiovascular diseases by inhibiting platelet aggregation. However, in future studies, the antiplatelet peptides containing PG(OG) sequences should be further verified by experiments, such as the drug affinity responsive target stability (DARTS) approach and Western blot, to further verify their target proteins.

## Figures and Tables

**Figure 1 foods-10-01553-f001:**
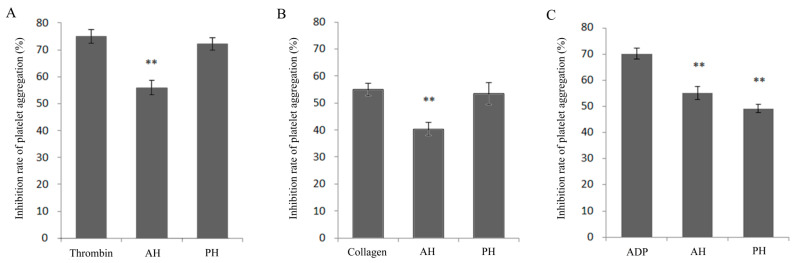
The platelet aggregation inhibition rate of Alcalase-hydrolysates (AH) and Protamex^®^-hydrolysates (PH) of fish skin collagen in different models induced by thrombin, collagen, and ADP. (**A**) Thrombin. (**B**) Collagen. (**C**) ADP. Mean ± SD, *n* = 5, ** *p* < 0.01, ANOVA, compared with the model group.

**Figure 2 foods-10-01553-f002:**
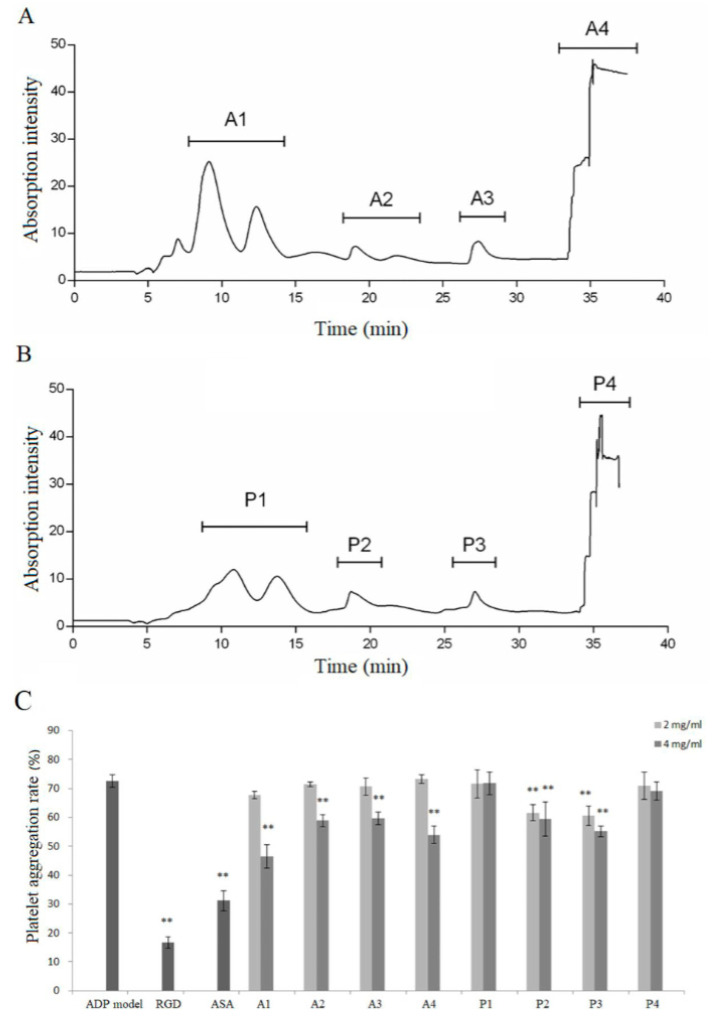
Separation scheme of antiplatelet peptides obtained from absorbate of skin collagen hydrolysates from *Hypophthalmichthys Molitrix* and their inhibition rate of ADP-induced platelet aggregation. (**A**,**B**) Reverse-phase chromatography on a C18 column. (**A**) Absorbates of Alcalase hydrolysates, AHA. (**B**) Absorbates of Protamex^®^ hydrolysates, PHA. Each sample was separated into 4 fractions, respectively (A1–A4; P1–P4). (**C**) Effect of each fraction on inhibiting ADP-induced platelet aggregation in vitro. Mean ± SD, *n* = 5, ** *p* < 0.01, ANOVA, compared with the model group.

**Figure 3 foods-10-01553-f003:**
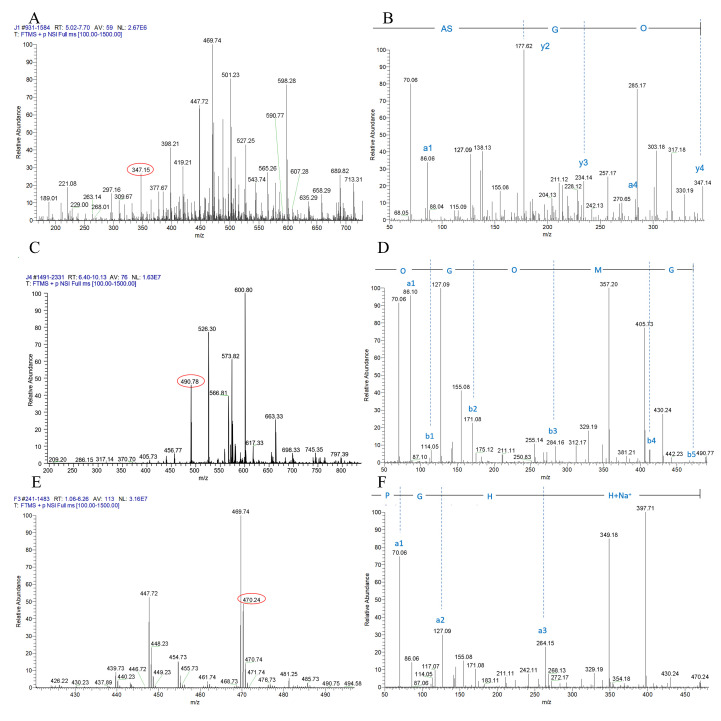
Identification of antiplatelet peptides from A1, A4, and P3. (**A**) Primary mass spectrum of A1. (**B**) Secondary mass spectrum of the peptide with *m*/*z* of 347.15. (**C**) Primary mass spectrum of A4. (**D**) Secondary mass spectrum of the peptide with *m*/*z* of 490.78. (**E**) Primary mass spectrum of P3. (**F**) Secondary mass spectrum of the peptide with *m*/*z* of 470.24.

**Figure 4 foods-10-01553-f004:**
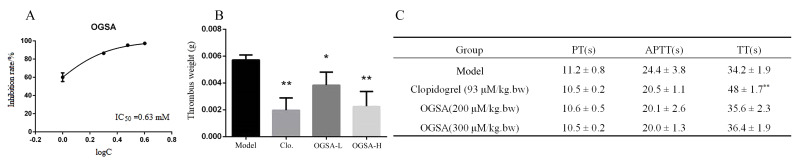
Evaluation of antiplatelet activity and side effects of OGSA. (**A**) Inhibition rate of OGSA on ADP-induced platelet aggregation in vitro. (**B**) The inhibitory effect of OGSA (200 and 300 μM/kg.bw) and the positive control Clopidogrel (93 μM/kg bw) on thrombus formation in vivo. (**C**) Effects of OGSA on PT, APTT, and TT in coagulation function test. Mean ± SD, *n* = 5, * *p* < 0.05, ** *p* < 0.01, ANOVA, compared with the model group.

**Figure 5 foods-10-01553-f005:**
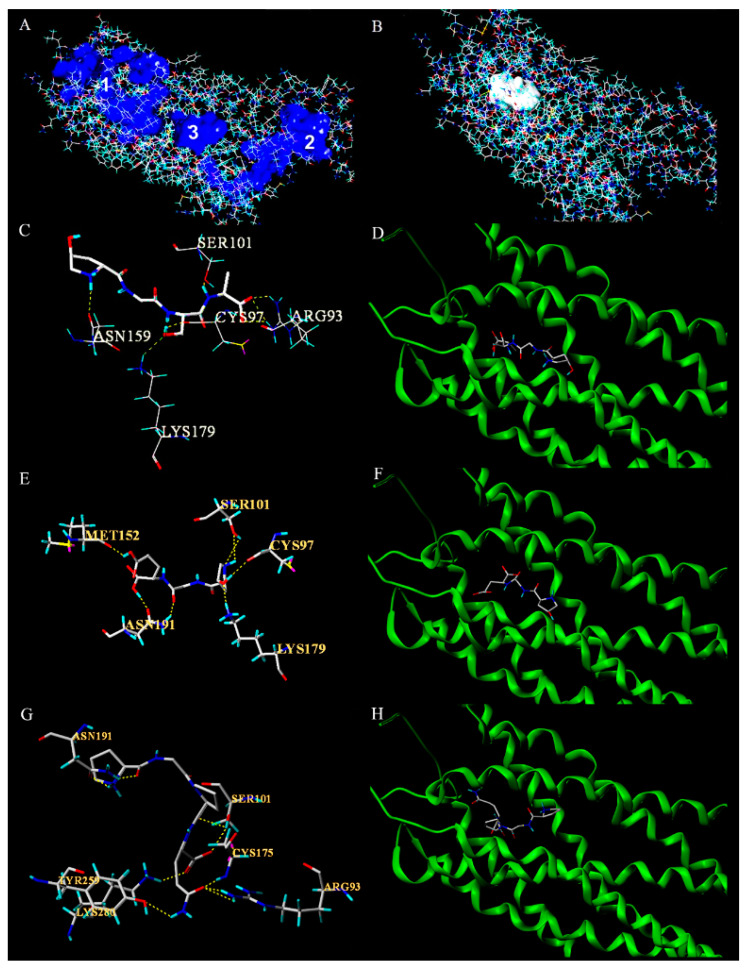
Molecular docking. (**A**) Three binding pockets on the surface of the P_2_Y_12_ receptor. (**B**) The pocket 1. (**C**) Docking of OGSA inside the receptor-binding site of P_2_Y_12_, showing hydrogen bonds with the catalytic sites (Asn159, Cys97, Ser101, Arg93, and Lys179). (**D**) The binding diagram of OGSA with P_2_Y_12_. (**E**) Docking of OGE inside the receptor-binding site of P_2_Y_12_, showing hydrogen bonds with the catalytic sites (Met152, Cys97, Ser101, Asn191, and Lys179). (**F**) The binding diagram of OGE with P_2_Y_12_. (**G**) Docking of PGPQ inside the receptor-binding site of P_2_Y_12_, showing hydrogen bonds with the catalytic sites (Arg 93, Cys175, Ser101, Asn191, Tyr259, and Lys280). (**H**) The binding diagram of PGPQ with P_2_Y_12_. The color of molecular diagram: White, C; Blue N; Aquamarine, H; Red, O; Yellow, S. Yellow dotted line: hydrogen bond.

**Table 1 foods-10-01553-t001:** Effect of 11 identified peptides on inhibiting platelet aggregation induced by different activators.

Sample	Peptide ^a^	Inhibition Rate of Platelet Aggregation %		
		Collagen Induced	Thrombin Induced	ADP ^b^ Induced
Control	RGD	17.1 ± 3.1	22.7 ± 2.7	79.7 ± 2.8
	Aspirin ^d^	55.0 ± 7.2	6.1 ± 2.5	58.4 ± 5.5
A1 ^c^	EGPAGPA	16.4 ± 2.2	53.9 ± 2.1	90.7 ± 4.4
	GTOGT	5.1 ± 27.7	−2.7 ± 7.1	88.0 ± 4.4
	PGPK	−18.7 ± 19.7	−2.4 ± 5.9	69.3 ± 0.5
	PGKP	−56.0 ± 10.7	0.1 ± 4.6	39.8 ± 1.9
	PGPQ	−51.2 ± 26.5	0.9 ± 1.0	64.5 ± 16.2
	PGQP	65.6 ± 14.1	−2.4 ± 2.9	56.2 ± 1.1
	OGSA	−0.1 ± 24.2	15.5 ± 4.4	96.4 ± 2.0
	OG	6.5 ± 0.7	−6.5 ± 2.2	56.6 ± 1.9
A4 ^c^	VVGOKG	−9.0 ± 0.9	11.5 ± 2.0	96.0 ± 1.7
	OGOMG	12.4 ± 19.4	0.3 ± 3.4	73.6 ± 7.4
P3 ^c^	PGHH	21.3 ± 8.7	19.6 ± 4.9	83.7 ± 3.5

^a^ The doses of peptides were 2 mM. ^b^ Adenosine Diphosphate, ADP. ^c^ A1, A4 and P3 are the components in Figure 2. ^d^ The dose of aspirin was 1.5 mM. Mean ± SD, *n* = 3.

**Table 2 foods-10-01553-t002:** The ΔG_bind_ binding energy (kcal/mol) of 13 antiplatelet peptides with P_2_Y_12_, PLC β2, and PLC β3, respectively.

No.	Sequence ^a^	IC_50_/mM ^b^	PLC β2 ^c^	PLC β3 ^c^	P_2_Y_12_ ^c^
1	OGE	0.62	−5.05	−5.24	−6.62
2	OGSA	0.63	−5.92	−3.88	−5.73
3	PGE	0.93	−5.22	−4.72	−6.09
4	PGEOG	0.97	−4.57	−4.44	−5.71
5	PGHH	1.4	−4.78	−4.33	−4.21
6	OGOMG	1.49	−4.55	−4.25	−3.93
7	PGPK	2.79	−6.24	−3.83	−3.27
8	OG	1.68	−6.11	−4.68	−4.68
9	PGKP	2.57	−4.6	−3.68	−2.92
10	VGPOGPA	2.73	−4.84	−3.11	−1.40
11	GTOGT	3.71	−2.00	−2.04	−1.46
12	PGPQ	3.56	−6.47	−5.66	−4.53
13	PGQP	2.31	−5.29	−5.21	−3.23
Pearson correlation coefficient			0.220	0.367	0.802

^a^ The antiplatelet peptides OGE, PGEOG, VGPOGP, and PGE were reported from a previous study [27]. ^b^ IC_50_ value: the half maximal inhibitory concentration. ^c^ PLC β2 and PLC β3 are the nodal proteins in the Gq-PLCβ signaling pathway. P_2_Y_12_ is the membrane surface receptor of ADP on the platelets.

## Data Availability

The data presented in this study are available on request from the corresponding author.

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
