# Peer review of "The Pro-Gly or Hyp-Gly Containing Peptides from Absorbates of Fish Skin Collagen Hydrolysates Inhibit Platelet Aggregation and Target P2Y12 Receptor by Molecular Docking"

_foods, 2021, doi:10.3390/foods10071553_

Round 1
Reviewer 1 Report
The manuscript from Tian et al. describes the identification and properties of peptides isolated from fish skin collagen, with an effect on platelet aggregation.
In my personal opinion, the manuscript is well written and the data presented support the claims and the conclusion. Therefore, it needs minor adaptations of the text.
I will however suggest some minor point to be considered to increase the readability of the text:
* Please define "O" as Hyp, as this letter is not straightforwardly associated with this special AA. Also consider that Hyp is only described in line 433;
* in my opinion, at first read the "Pro(Hyp)-Gly" sounds a bit confusing. I suggest the author to use a more unambiguous (Pro-Gly and Hyp-Gly) in the title;
* in analogy to the previous comments I think it is best to define "PG(OG)" in the main text as well as in the abstract (e.g. "Pro-Gly and Hyp-Gly (PG(OG))" );
* please uniform the nomenclature of the sequence as in the manuscript you can read: Pro(Hyp)-Gly (which could be P(O)G), OG(PG), and PG(OG);
* in material and methods "hypophthalmichthys molitrix" should be italicized;
* please define TEER (line 112);
* lines 283-284: "According to the results," should be revised with "According to platelet aggregation inhibition results,";
* figure 5: the figure is a bit difficult to read, I suggest to the authors to use a white background and change the other colors to enhance contrast;
* line 423: as the binding residues are identified via docking studies I will replace the stronge "are" in "binding sites that improve the activity are" with a milder "might be". Moreover, these residues are responsible for the recognition of the PG(OG) peptides, not for the improvement of their activity. Please correct.
Author Response
Point 1: Please define "O" as Hyp, as this letter is not straightforwardly associated with this special AA. Also consider that Hyp is only described in line 433. 

Response 1: The definition of “O” has been added in abstract (see L16).
Point 2: In my opinion, at first read the "Pro(Hyp)-Gly" sounds a bit confusing. I suggest the author to use a more unambiguous (Pro-Gly and Hyp-Gly) in the title.
Response 2: This description has been revised in the title. “The Pro-Gly or Hyp-Gly containing peptides from absorbates of fish skin collagen hydrolysates inhibit platelet aggregation and target P2Y12 receptor by molecular docking”
Point 3: In analogy to the previous comments I think it is best to define "PG(OG)" in the main text as well as in the abstract (e.g. "Pro-Gly and Hyp-Gly (PG(OG))" ).
Response 3: This description has been revised in the abstract.
“Nine of them contained Pro-Gly and Hyp-Gly (PG(OG)) sequence...” (see L16).
Point 4: Please uniform the nomenclature of the sequence as in the manuscript you can read: Pro(Hyp)-Gly (which could be P(O)G), OG(PG), and PG(OG).
Response 4: All the nomenclature of the sequence Pro-Gly and Hyp-Gly has been uniformed as PG(OG) (see L349, 359 & 370).
Point 5: in material and methods "hypophthalmichthys molitrix" should be italicized.
Response 5: These words “hypophthalmichthys molitrix” have been italicized (see L84 & 98).
Point 6: Please define TEER (line 112).
Response 6: The definition of “Trans-epithelial electrical resistance (TEER)” has been added (see L121).
Point 7: Lines 283-284: "According to the results," should be revised with "According to platelet aggregation inhibition results,".
Response 7: This phrase has been revised. “According to platelet aggregation inhibition results (Table 1 and Figure S1)...” (see L298).
Point 8: Figure 5: the figure is a bit difficult to read, I suggest to the authors to use a white background and change the other colors to enhance contrast;
Response 8: Since the hydrogen bonds in the pictures exported by the software are yellow, the use of a white background will make the hydrogen bonds difficult to identify, so most studies used black as the background of the molecular docking pictures.
In Figures 5D, 5F and 5H, We have changed the gray P2Y12 molecular model to green to make the reader identify it more clearly. Besides, we have enhanced the contrast and color saturation of the picture to make it clearer. High-resolution pictures have been uploaded to the picture attachment.
Point 9: Line 423: as the binding residues are identified via docking studies I will replace the stronge "are" in "binding sites that improve the activity are" with a milder "might be". Moreover, these residues are responsible for the recognition of the PG(OG) peptides, not for the improvement of their activity. Please correct.
Response 9: This mistake phrase has been corrected. “...we found that the key binding sites that responsible for the recognition of the PG(OG) peptides might be Cys 97, Lys 179 and Ser 101.” (see L443).
We have replaced “are” with “might be”. “The key binding sites in pocket 1 of P2Y12 receptor with PG(OG) might be Cys97, Ser101 and Lys179...” (see L467).
Reviewer 2 Report
The manuscript The Pro(Hyp)-Gly-containing peptides from absorbates of fish skin collagen hydrolysates inhibit platelet aggregation and tar-3 get P2Y12 receptor by molecular docking explores the antiplatelet potential of peptides obtained from silver carp skin collagen.
The authors assess this activity by in vitro and in vivo tests, after simulated digestion and absortion of peptides, and provide the mode of action by computer assisted molecular docking.
The methods used were appropriate for the aim of the work and the several different approaches (in vivo, in vitro, sequencing and computing) strongly supports the results presented, together with previous work published by the same authors.
Since there are limited studies that evaluate the potential asset of fish collagen proteins as human food additives I believe that this manuscript will be relevant for the field.
For these reasons I recommend a few amendments, detailed below:
Abstract
Previous studies found that the collagen hydrolysates of fish skin have antiplatelet activity. The unknown antiplatelet component in that needs to be revealed.
I suggest rephrase this as: Previous studies found that the collagen hydrolysates of fish skin have antiplatelet activity but this component remained unknown.
Protamex
I think you should write Protamex® here and throughout the text.
Hypophthalmichthys Molitrix
Hypophthalmichthys molitrix here and throughout the text.
Introduction
L44-45 Compared with antiplatelet drugs, to prevent and suppress the thrombotic diseases through diets has its unique advantages:
Rephrase. Suggestion: Comparing with aniplatelet drugs, preventing and suppressing thrombotic diseases through diets has unique advantages:
L51-52 In recent years, the separation and identification of bioactive peptides from collagen hydrolysates has attracted the attention of researchers, such as antioxidant peptides.
Rephrase. Suggestion: In recent years, the separation and identification of bioactive peptides from collagen hydrolysates, such as antioxidant peptides, has attracted the attention of researchers.
L57 Silver carp
You should use here the scientific name as well, since is the first time you mention it in the text. So: …..silver carp (Hypophthalmichthys molitrix)… Then, you can use H. molitrix or just the common name.
L62-63 In this study, we used Alcalase and Protamex respectively instead of the previous mixed protease (Papain, Trypsin, Alcalase) to prepare collagen.
This phrase seems more suitable for material and methods section.
L63-64 Considering that collagen peptides were taken orally, the target of this research was the absorbates of collagen hydrolysates.
You say basically the same in lines 71-72, I would delete this phrase, just to avoid repetition.
Materials and Methods
L109 digest the cells using trypsinase containing 0.5% EDTA
I think you mean dislodge cells using trypsine.
L112 After cultivating 21 days, the TEER value is 800 or higher, Ω cm2, shows that the cells differentiated into monolayer cells
This sentence is a bit confusing to me. Maybe you could rephrase it as: After 21 days of incubation, a TEER value of 800 Ω cm2 or higher indicated that the cells were differentiated in monolayers.
L116-117 Then placed the cell model in a cell incubator and balanced at 37 ℃ for 30 min.
Then, the cell model was placed….
Author Response
Point 1: “Previous studies found that the collagen hydrolysates of fish skin have antiplatelet activity. The unknown antiplatelet component in that needs to be revealed.”
I suggest rephrase this as: Previous studies found that the collagen hydrolysates of fish skin have antiplatelet activity but this component remained unknown. 

Response 1: Thank you for your suggestion. This phrase has been revised in abstract.
“Previous studies found that the collagen hydrolysates of fish skin have antiplatelet activity but this component remained unknown.” (see L12-13)
Point 2: I think you should write Protamex® here and throughout the text.
Response 2: All “Protamex” in the text has been modified to “Protamex®”.
Point 3: Hypophthalmichthys molitrix here and throughout the text.
Response 3: These words “hypophthalmichthys molitrix” have been italicized (see L84 & 98).
Point 4: L44-45 Compared with antiplatelet drugs, to prevent and suppress the thrombotic diseases through diets has its unique advantages:
Rephrase. Suggestion: Comparing with aniplatelet drugs, preventing and suppressing thrombotic diseases through diets has unique advantages:
Response 4: Thank you for your suggestion. This phrase has been revised.
“Comparing with aniplatelet drugs, preventing and suppressing thrombotic diseases through diets has unique advantages: ” (see L45-46).
Point 5: L51-52 In recent years, the separation and identification of bioactive peptides from collagen hydrolysates has attracted the attention of researchers, such as antioxidant peptides.
Rephrase. Suggestion: In recent years, the separation and identification of bioactive peptides from collagen hydrolysates, such as antioxidant peptides, has attracted the attention of researchers.
Response 5: Thank you for your suggestion. This sentence has been revised.
“In recent years, the separation and identification of bioactive peptides from collagen hydrolysates, such as antioxidant peptides [19,20], anti-inflammatory peptides [21] and anti-osteoporosis peptides [22], has attracted the attention of researchers.” (see L52-54).
Point 6: L57 Silver carp. You should use here the scientific name as well, since is the first time you mention it in the text. So: …..silver carp (Hypophthalmichthys molitrix)… Then, you can use H. molitrix or just the common name..
Response 6: The words “Silver carp” has been replaced with “Hypophthalmichthys Molitrix” (see L61 & 65).
Point 7: L62-63 In this study, we used Alcalase and Protamex respectively instead of the previous mixed protease (Papain, Trypsin, Alcalase) to prepare collagen.
This phrase seems more suitable for material and methods section.
Response 7: This phrase has been removed. We thought this description is suitable for discussion and rephrased it. “In this study, Alcalase and Protamex® were respectively used to prepare collagen hydrolysates instead of the previous mixed enzymes.” (see L403-405).
Point 8: L63-64 Considering that collagen peptides were taken orally, the target of this research was the absorbates of collagen hydrolysates.
You say basically the same in lines 71-72, I would delete this phrase, just to avoid repetition.
Response 8: This phrase is indeed similar to the content that follows (L77-79) and it has been deleted.
Point 9: L109 digest the cells using trypsinase containing 0.5% EDTA
I think you mean dislodge cells using trypsine.
Response 9: This was a wrong description and it has been corrected.
“When Caco-2 cells covered 80% to 90% of the culture area of the flask, dislodged cells using trypsin.” (see L117).
Point 10: L112 After cultivating 21 days, the TEER value is 800 or higher, Ω cm2, shows that the cells differentiated into monolayer cells
This sentence is a bit confusing to me. Maybe you could rephrase it as: After 21 days of incubation, a TEER value of 800 Ω cm2 or higher indicated that the cells were differentiated in monolayers.
Response 10: Thanks for your suggestion. This confusing description has been revised.
“After 21 days of incubation, a Trans-epithelial electrical resistance (TEER) value of 800 Ω cm2 or higher, indicated that the cells were differentiated in monolayers.” (see L120-123).
Point 11: L116-117 Then placed the cell model in a cell incubator and balanced at 37 ℃ for 30 min.
Then, the cell model was placed….
Response 11: This misleading description has been revised.
“Then, the cell model was placed in a cell incubator and balanced at 37 ℃ for 30 min.” (see L127).
This manuscript is a resubmission of an earlier submission. The following is a list of the peer review reports and author responses from that submission.